# Enhanced Orientation Tracking for Redundant Manipulators via DNN-Based Double Control

Chenrui Xu
*School of Information Science and Engineering*
*Lanzhou University*
Lanzhou, China
320220939681@lzu.edu.cn

Yuheng Qian
*School of Information Science and Engineering*
*Lanzhou University*
Lanzhou, China
qianyh21@lzu.edu.cn

*Abstract*—This paper proposes a new double-indicator control scheme of redundant manipulators, which utilizes an untrained dynamic neural network (DNN) solver. The control scheme combines direction tracking, physical constraints, and anti-noise design to address problems of the high computational complexity and the lack of direction tracking in existing neural network-based solutions. In addition, the DNN solver provides a control-theoretic framework which ensures the global and exponential convergence, stability, and robustness of the control scheme. In our design, we specifically consider the effect of noises on the system and incorporate the anti-noise mechanism. Furthermore, the effectiveness and feasibility of the proposed control scheme are verified through simulations with a KUKA LBR iiwa 7 R800 manipulator. The results show that the DNN-based double-indicator control scheme can efficiently generate accurate motion trajectories while maintaining the directional stability of the end-effector, and can resist noises.

*Index Terms*—dynamic neural network (DNN), double control, redundant manipulators, orientation maintenance, noise tolerance

## I. INTRODUCTION

As non-redundant manipulators face difficulties in adapting to increasingly complex work environments such as precision manufacturing, medical surgery, and space exploration, redundant manipulators with seven or more degrees of freedom (DOF) have been widely used in various fields due to their high flexibility and reliability [1], [2]. However, redundant manipulators' significant control complexity and computational requirements limit the application in cost-sensitive environments and hinder further popularization [3]. Existing control schemes typically require significant computational resources and have real-time performance and robustness limitations. In addition, conventional control schemes often exhibit deficiencies in adaptability and stability when dealing with noises and physical constraints such as joint and speed limits [4], [5]. Therefore, developing an efficient, stable, and noise-resistant control scheme that can accurately control the redundant manipulator in complex environments has become a key research priority.

A fundamental and essential problem in controlling the redundant manipulator is the solution of the motion planning and inverse kinematics problems [6], [7]. The redundant manipulator has redundant degrees of freedom, resulting in multiple possible combinations of solutions for the joint angle at the same end position and posture [8]. In addition, the physical properties of the redundant manipulator and other objectives (e.g., optimizing energy consumption, improving workspace efficiency) are also factors to be considered in the control scheme [9]. In control schemes of the redundant manipulator, pseudo-inverse methods are common means to solve the problem of inverse kinematics [10]. For the manipulator with the specific structure (e.g., spherical wrist structure, 6-DOF manipulators [11]), the pseudo-inverse method is fast and accurate [12], [13]. However, due to the physical limitations of the manipulator (e.g., joint angle range and joint velocity limitations [4], [5]), the pseudo-inverse method often fails to account for these limitations adequately. At the same time, they are difficult to optimize for additional objectives [14]. Therefore, the pseudo-inverse method cannot be used alone in practical applications of the redundant manipulator [15].

In order to satisfy the desired objective, more sophisticated optimization methods such as quadratic programming (QP) are usually required [16]. The QP method transforms joint constraints and other physical limitations into inequality constraints versus equational constraints, thus allowing the accommodation of these constraints when solving inverse kinematics problems [17]. Currently, redundant manipulator control strategies have achieved a wide range of applications based on optimization solution methods, including the minimum kinetic energy (MKE) scheme and the cyclic motion generation (CMG) scheme [18].

The MKE scheme aims to minimize the kinetic energy of the manipulator by optimizing joint speeds, resulting in low power consumption and smooth motion. The scheme is well suited for long-term continuous operation or energy-sensitive applications but may not consider physical constraints such as joint angle ranges and velocity limits. In addition, it ignores control objectives such as motion accuracy and stability [19].

The CMG scheme is a method that improves the motion accuracy and stability of the redundant manipulator by designing cyclic motion trajectories [20]. The method corrects the joint drift phenomenon and thus improves the repeatability and accuracy of the motion. However, it may face high computational complexity during the optimization process and

may not directly optimize energy consumption, which limits its applicability in energy-sensitive applications. In this paper, we propose a new double-indicator control scheme that fully combines the advantages of two control schemes to improve the accuracy and smoothness of the motion of the redundant manipulator.

In addition, much research has been devoted to applying neural networks in the control of the manipulator, and they are well supported by theory [21]–[23]. In [24], a conventional adaptive neural network (NN) controller is optimized to achieve global stability by introducing the switching mechanism. Furthermore, recurrent neural networks (RNN) have unique advantages in controlling the manipulator [25]–[27]. The hidden layer of RNN takes the hidden state of the previous step as input and decides the current hidden state along with the current input, which makes the RNN highly effective in dealing with time series, natural language processing, and other tasks involving sequential information [28], [29]. In [30], an RMG scheme based on orthogonal projection (OPRMG) is presented, and a novel RNN is constructed by the gradient descent method and gradient compensation, theoretically eliminating the position error. By adding more dynamic adjustment mechanisms, DNN not only inherits the ability of RNN to process sequence data but also adjusts the structure or parameters according to the dynamic changes of the input data, which enhances flexibility and adaptability. In addition, DNN also realizes the simultaneous processing of multiple related tasks [31]. Furthermore, noise is unavoidable in the actual control system [32], which may originate from various aspects such as the control signal, the precision limitation of hardware implementation, and other factors, including environmental disturbances and sensor inaccuracies [33]. However, when constructing RNN for the real-time control computation, it is often assumed that there is no system or external noises [34], [35]. However, there are diverse methods to deal with noise from the traditional control perspective, such as integral control and other complex internal model control. In [36], the design and implementation of DNN for the kinematic force control of the manipulator is discussed under polynomial noises, and an improved FP-DNN solver is presented to enhance the robustness of the system effectively eliminates the effect of polynomial noises by automatically adjusting the convergence parameters. In addition, in terms of error control, a novel zeroed neural network (ZNN) model is presented in [37] for solving the problem of cyclic motion planning (CMP) for the redundant manipulator under physical constraints, and it achieves the improvement of the accuracy of the control with the QP-based CMP scheme. The paper further proposes an untrained DNN solver that ensures global convergence, stability, and noise resistance and significantly reduces computational complexity, providing a more efficient and robust solution for controlling redundant manipulators. The paper also enhances noise control capabilities by introducing a simple noise-resistant mechanism, building on the dual-criteria approach.

The main contributions of this paper are (1) a novel double-indicator control framework that combines kinetic energy minimization and cyclic motion generation to improve the smoothness and accuracy of manipulator motion significantly; (2) the introduction of an untrained dynamic neural network solver that ensures global convergence, stability, and noise resistance while reducing computational complexity; (3) the development of a simple sliding average filter as a noise reduction mechanism, which enhances the system's robustness in noisy environments; and (4) validation of the proposed control scheme through simulations with the KUKA LBR iiwa 7 R800 manipulator, demonstrating the method's effectiveness and feasibility.

The rest of the paper is divided into four parts. In Section II, the joint angle range limit, angular velocity limit, and other physical factors of the redundant manipulator are transformed into constraints of the optimization problem, the objective function is constructed based on the double-indicator scheme, and the end-effector posture is considered. The solver of the dynamic neural network is constructed, and its global exponential convergence is analyzed in Section III. In Section IV, random errors in joint angular velocities are introduced to simulate unavoidable systematic errors in the actual control process, and the simple sliding average anti-noise approach effectively reduces the effect of the noise. In Section V, results of simulations using 7-DOF KUKA LBR iiwa 7 R800 manipulators are given to compare the performance of the control schemes.

## II. Optimising Problem Construction

This section presents the double-indicator control scheme that considers both the minimum kinetic energy objective and the objective of cyclic motion generation, constitutes an optimized objective function, and transforms the original problem into a quadratic programming problem by regrading the physical constraints of joints as equation or inequality constraints.

For a manipulator with $k$ degrees of freedom, its transition from the joint angle space $\boldsymbol{\theta}(t) \in \mathbb{R}^k$ to the end-effector state space $\boldsymbol{p}(t) \in \mathbb{R}^b$ is realized in forward kinematics by the mapping function [38], [39]:

$$\boldsymbol{\theta}(t) = \boldsymbol{\psi}(\boldsymbol{p}(t)), \tag{1}$$

where $\boldsymbol{\psi}(\cdot)$ contains the physical information of the manipulator, such as the joint offset, linkage length, linkage torsion angle, and other relevant parameters, and $b$ is the dimension of the manipulator's workspace. We can illustrate this in detail by using the denavit-hartenberg (DH) parameter and the transformation matrix of the manipulator. The transformation matrix is viewed as the following structure:

$$A_i = \begin{bmatrix} R_i & \boldsymbol{l}_i \\ 0 & 1 \end{bmatrix},$$

where $R_i \in \mathbb{R}^{3 \times 3}$, is a rotation matrix and $\boldsymbol{l}_i \in \mathbb{R}^3$ is a translation vector. For the joint angle space $\boldsymbol{\theta}(t) \in \mathbb{R}^k$, the total transformation matrix is written as

$$T = A_1 \cdot A_2 \cdot A_3 \cdot \cdots \cdot A_k.$$

The spatial coordinate of the end-effector relative to the origin on the base coordinate system is written as

$$\boldsymbol{p} = T(1:3,4).$$

By deriving both sides of (1) with respect to time, the mapping relation for the velocity is obtained, thereby constructing a concise and efficient affine system:

$$\dot{\boldsymbol{p}}(t) = J\dot{\boldsymbol{\theta}}(t), \tag{2}$$

where $\dot{\boldsymbol{p}}(t)$ is the velocity of the end-effector; $J = \partial \psi(\boldsymbol{\theta})/\partial \boldsymbol{\theta} \in \mathbb{R}^{b \times k}$ denotes the Jacobi matrix; The $\dot{\boldsymbol{\theta}}(t)$ denotes the joint velocity.

Furthermore, the posture of the end-effector of a redundant manipulator with more than 6 DOF may be shifted in 3D space. Thus, the posture needs to be controlled as well. The posture of the end-effector can be obtained by the rotating matrix description:

$$R = \begin{bmatrix} x_1 & y_1 & z_1 \\ x_2 & y_2 & z_2 \\ x_3 & y_3 & z_3 \end{bmatrix}. \tag{3}$$

Since the rotation matrix $R$ is an orthogonal matrix with determinant 1, any second-order submatrix of it uniquely determines the entire matrix. Therefore, we reconstruct the direction vectors as $\boldsymbol{r} = \begin{bmatrix} x_1 & y_1 & x_2 & y_2 \end{bmatrix}^{\mathrm{T}}$. Thus, the control of the posture is described as $\boldsymbol{r}(t) = \boldsymbol{r}(0)$, and the dynamic control equation is constructed as

$$\dot{\boldsymbol{r}}(t) - \dot{\boldsymbol{r}}_d(t) = -\nu(\boldsymbol{r}(t) - \boldsymbol{r}_d(t)), \tag{4}$$

where $\nu > 0$ is the error amplification factor with respect to the posture of the end-effector. By introducing the Jacobian matrix $J' = \partial \boldsymbol{r}/\partial \theta \in \mathbb{R}^{4 \times k}$ to $\dot{\boldsymbol{r}}(t) = J'\dot{\theta}$, we get:

$$J'\dot{\boldsymbol{\theta}} = \dot{\boldsymbol{r}}_d - \nu(\boldsymbol{r} - \boldsymbol{r}_d). \tag{5}$$

Since the solution of (2) is not unique for $k > 6$, a reasonable optimisation objective function is specified by constructing a double-indicator scheme to determine a solution that meets the objective, satisfies the physical constraints and geometrical structure, and restricts the end-effector posture. Since equations in the following section are mostly dynamic equations and most of the variables are time series, the time variable $t$ is removed to simplify the description without affecting the understanding. Then the double-indicator control scheme is described as follows:

$$\min_{\dot{\boldsymbol{\theta}}} \quad \frac{1}{2}\left(\xi\dot{\boldsymbol{\theta}}^T G(\boldsymbol{\theta})\dot{\boldsymbol{\theta}} + (1-\xi)\|\dot{\boldsymbol{\theta}} + m(\boldsymbol{\theta} - \boldsymbol{\theta}_0)\|_2^2\right)$$
$$\text{s.t.} \quad J\dot{\boldsymbol{\theta}} = \dot{\boldsymbol{p}}_d$$
$$J'\dot{\boldsymbol{\theta}} = \dot{\boldsymbol{r}}_d$$
$$\lambda_i \le \dot{\theta}_i \le \lambda_i^+, \quad i = 1, 2, \ldots, k, \tag{6}$$

where $\xi \in [0, 1]$ is used as the weight coefficients of different performance indicators; $\|\cdot\|_2$ denotes the Euclidean paradigm

of a vector or matrix; $\boldsymbol{\theta}_0$ is the vector consisting of the initial angles of each joint of the redundant manipulators; $m > 0$ is the moderating factor of the CMG scheme; $\lambda_i^+ = \min\left\{\dot{\overline{\theta}}_i, \kappa(\overline{\theta_i} - \theta_i)\right\}$, $\lambda_i^- = \max\left\{\dot{\underline{\theta}}_i, \kappa(\underline{\theta}_i - \theta_i)\right\}$ where $\dot{\overline{\theta}}_i$ and $\dot{\underline{\theta}}_i$ denote the upper and lower limits of the velocity of the $i$th joint, respectively, and the upper and lower limits of the joint angle are defined in the same way. Since the limitation of the size of the joint angle adjustment needs to be considered at the same time, $\kappa(\underline{\theta}_i - \theta_i)$ is also taken into account, and $\kappa$ is used as a coefficient to appropriately scale or adjust the joint angle when it exceeds the permissible range. For subsequent ease of presentation, this optimization problem is formally simplified below. Expand and simplify the objective function:

$$\min_{\dot{\boldsymbol{\theta}}} \frac{1}{2}\xi\dot{\boldsymbol{\theta}}^T G(\boldsymbol{\theta})\dot{\boldsymbol{\theta}} + \frac{1}{2}(1-\xi)\|\dot{\boldsymbol{\theta}} + m(\boldsymbol{\theta} - \boldsymbol{\theta}_0)\|_2^2.$$

After expanding the quadratic term and ignoring the constant term, it is written as:

$$\min_{\dot{\boldsymbol{\theta}}} \frac{1}{2}\xi\dot{\boldsymbol{\theta}}^T G(\boldsymbol{\theta})\dot{\boldsymbol{\theta}} + \frac{1}{2}(1-\xi)\left(\dot{\boldsymbol{\theta}}^T\dot{\boldsymbol{\theta}} + 2m\dot{\boldsymbol{\theta}}^T(\boldsymbol{\theta} - \boldsymbol{\theta}_0)\right).$$

Definition: $\Phi = \xi G(\boldsymbol{\theta}) + (1-\xi)I$, $\eta = (1-\xi)m(\boldsymbol{\theta} - \boldsymbol{\theta}_0)$. Thus, the objective function is written as:

$$\min_{\dot{\boldsymbol{\theta}}} \frac{1}{2}\dot{\boldsymbol{\theta}}^T \Phi\dot{\boldsymbol{\theta}} + \boldsymbol{\eta}^T\dot{\boldsymbol{\theta}}.$$

By reconstructing the matrix and vectors, we obtain: $\Phi = \boldsymbol{\xi}G(\boldsymbol{\theta}) + (1-\boldsymbol{\xi})\boldsymbol{I} \in \mathbb{R}^{k \times k}$, $\boldsymbol{\xi} = [J; J'] \in \mathbb{R}^{(b+4) \times k}$, $\mathbf{d} = [\dot{\boldsymbol{p}}_d; \dot{\boldsymbol{r}}_d] \in \mathbb{R}^{b+4}$, $\boldsymbol{\eta} = (1-\boldsymbol{\xi})m(\boldsymbol{\theta} - \boldsymbol{\theta}_0) \in \mathbb{R}^k$, $\boldsymbol{\Omega} = [\boldsymbol{I}; -\boldsymbol{I}] \in \mathbb{R}^{(2k) \times k}$, $\boldsymbol{\epsilon} = \dot{\boldsymbol{\theta}} \in \mathbb{R}^k$, $\boldsymbol{\chi} = [\boldsymbol{\lambda}^+; \boldsymbol{\lambda}^-] \in \mathbb{R}^{2k}$. The identity matrix $\boldsymbol{I} \in \mathbb{R}^{k \times k}$, $\boldsymbol{\lambda}^- \le \boldsymbol{\epsilon} \le \boldsymbol{\lambda}^+$, where $\boldsymbol{\lambda}^-$ ($\boldsymbol{\lambda}^+$) represents the lower (upper) limit of the joint velocity. The suggested double-indicator control technique is rewritten as

$$\min_{\boldsymbol{\epsilon}} \quad \frac{1}{2}\boldsymbol{\epsilon}^T \Phi\boldsymbol{\epsilon} + \boldsymbol{\eta}^T\boldsymbol{\epsilon}$$
$$\text{s.t.} \quad \boldsymbol{\xi}\boldsymbol{\epsilon} = \mathbf{d}$$
$$\boldsymbol{\Omega}\boldsymbol{\epsilon} \le \boldsymbol{\chi}, \tag{7}$$

which is a QP problem with $\boldsymbol{\epsilon}$ as the decision variable.

## III. DYNAMIC NEURAL NETWORK SOLVER

This section describes the construction of the proposed dynamic neural network solver (6) and provides proofs of the global convergence and exponential convergence. Unlike traditional neural networks, the untrained dynamic neural network (DNN) solver proposed in this paper offers several key advantages: (1) it does not require pre-training, allowing it to adapt flexibly to real-time control without the need for extensive datasets; (2) it ensures global convergence and stability, even in complex multi-degree-of-freedom control tasks, making it robust against varying environmental conditions; (3) it has inherent noise-resistance capabilities, enhancing its effectiveness in noisy environments; and (4) it reduces computational complexity, enabling more efficient real-time

control, particularly in resource-constrained systems. These properties make the DNN solver highly suitable for controlling redundant manipulators under challenging conditions.

To obtain the global optimal solution of the convex QP problem (7), the KKT conditions need to be satisfied [40], [41]. The Lagrangian function is constructed as follows:

$$\boldsymbol{L}(\boldsymbol{\epsilon}, \boldsymbol{\nu}, \boldsymbol{\mu}) = \frac{1}{2}\boldsymbol{\epsilon}^T\Phi\boldsymbol{\epsilon}+\boldsymbol{\eta}^T\boldsymbol{\epsilon}+\boldsymbol{\nu}^T(\boldsymbol{\xi}\boldsymbol{\epsilon}-\mathbf{d})+\boldsymbol{\mu}^T(\Omega\boldsymbol{\epsilon}-\boldsymbol{\chi}), \quad (8)$$

where $\boldsymbol{\nu} \in \mathbb{R}^{b+4}$ and $\boldsymbol{\mu} \in \mathbb{R}^{2k}$ are KKT multipliers.

The following KKT conditions need to be met in order to solve the optimization problem:

$$\frac{\partial \boldsymbol{L}}{\partial \boldsymbol{\epsilon}} = \Phi\boldsymbol{\epsilon} + \boldsymbol{\eta} + \boldsymbol{\xi}^T\boldsymbol{\nu} + \Omega^T\boldsymbol{\mu} = \mathbf{0},$$
$$\frac{\partial \boldsymbol{L}}{\partial \boldsymbol{\nu}} = \boldsymbol{\xi}\boldsymbol{\epsilon} - \mathbf{d} = \mathbf{0},$$
$$\frac{\partial \boldsymbol{L}}{\partial \boldsymbol{\mu}} = \Omega\boldsymbol{\epsilon} - \boldsymbol{\chi} = \mathbf{0},$$
$$\underline{\boldsymbol{\epsilon}} \le \boldsymbol{\epsilon} \le \overline{\boldsymbol{\epsilon}},$$
$$\mu_i \ge 0, \quad i = 1, 2, \ldots, 2k,$$

where $\mu_i$ denotes the $i$th element of $\mu$; $\overline{\epsilon}$ and $\underline{\epsilon}$ are the upper and lower bounds of $\boldsymbol{\epsilon}$, respectively. Their $i$th elements are:

$$\overline{\epsilon_i} = \min\{\overline{\dot{\theta}_i}, \kappa(\overline{\theta_i} - \theta_i)\}, \quad i = 1, 2, \ldots, k$$
$$\underline{\epsilon_i} = \max\{\underline{\dot{\theta}_i}, \kappa(\underline{\theta_i} - \theta_i)\}, \quad i = 1, 2, \ldots, k.$$

To effectively handle inequality constraints in (7), a DNN solver is constructed in the following form:

$$\rho\dot{\boldsymbol{\epsilon}} = -\boldsymbol{\epsilon} + \Pi_{\boldsymbol{\epsilon}}(\boldsymbol{\epsilon} - (\Phi\boldsymbol{\epsilon} + \boldsymbol{\eta} + \boldsymbol{\xi}^T\boldsymbol{\nu} + \Omega^T\boldsymbol{\mu})),$$
$$\rho\dot{\boldsymbol{\nu}} = -\boldsymbol{\nu} + \Pi_{\boldsymbol{\nu}}(\boldsymbol{\nu} + \boldsymbol{\xi}\boldsymbol{\epsilon} - \mathbf{d}),$$
$$\rho\dot{\boldsymbol{\mu}} = -\boldsymbol{\mu} + \Pi_{\boldsymbol{\mu}}(\boldsymbol{\mu} + \Omega\boldsymbol{\epsilon} - \boldsymbol{\chi}),$$

where the convergence coefficient $\rho > 0$. Additionally, in the DNN solver (7), projection functions $\Pi_{\boldsymbol{\epsilon}}$, $\Pi_{\boldsymbol{\nu}}$, and $\Pi_{\boldsymbol{\mu}}$ are used to enforce range constraints on the variables $\boldsymbol{\epsilon}$, $\boldsymbol{\nu}$, and $\boldsymbol{\mu}$. They are defined as follows:

$$\Pi_{\boldsymbol{\epsilon}}(\epsilon_i) = \begin{cases} \overline{\epsilon_i}, & \epsilon_i \ge \overline{\epsilon_i} \\ \epsilon_i, & \underline{\epsilon_i} < \epsilon_i < \overline{\epsilon_i}, \\ \underline{\epsilon_i}, & \epsilon_i \le \underline{\epsilon_i} \end{cases}$$

$$\Pi_{\boldsymbol{\nu}}(\nu_i) = \nu_i,$$

$$\Pi_{\boldsymbol{\mu}}(\mu_i) = \begin{cases} \mu_i, & \mu_i \ge 0 \\ 0, & \mu_i < 0. \end{cases}$$

## IV. NOISE MODELLING AND INTRODUCTION OF ANTI-NOISE MECHANISMS

This section focuses on modeling the system noise during the control of the manipulator and incorporating additional anti-noise mechanisms.

Gaussian white noise is used to imitate genuine noise conditions because it considers the variability and instability of the noise in real-world scenarios. The angular velocity $\dot{\boldsymbol{\theta}}$ of the joint of the manipulator is mainly impacted by the noise in the

trajectory control of a manipulator's end effector. Simplifying the control system:

$$\dot{\boldsymbol{x}}(t) = \boldsymbol{f}(\boldsymbol{x}(t), \boldsymbol{u}(t)),$$

where the state vector of the control system is $\boldsymbol{x}(t)$, and $\boldsymbol{u}(t)$ is the control input, including the joint velocity vector $\boldsymbol{w}(t)$. The control algorithm for the joint velocity is expressed as:

$$\boldsymbol{w}(t) = \boldsymbol{k}(\boldsymbol{x}(t)),$$

where $\boldsymbol{k}$ is an abstract function that computes the joint velocity. To simulate random errors in the manipulator or the environment, the noise is added to the computed joint velocity $\boldsymbol{w}(t)$:

$$\boldsymbol{w}_{\text{noisy}}(t) = \boldsymbol{w}(t) + \boldsymbol{n}(t),$$

where $\boldsymbol{n}(t)$ is the noise vector, typically assumed to be zero-mean Gaussian white noise:

$$\boldsymbol{n}(t) \sim \mathcal{N}(0, \sigma^2).$$

A simple moving average filter is introduced to smooth the noisy joint velocity:

$$\boldsymbol{w}_{\text{filtered}}(t) = \frac{1}{N} \sum_{i=0}^{N-1} \boldsymbol{w}_{\text{noisy}}(t - i\Delta t), \quad (9)$$

where $N$ is the size of the moving window, and $\Delta t$ is the time step. At each time step, the smoothed joint angular velocity $\mathbf{w}_{\text{filtered}}(t)$ is used as part of the state vector to update the system state:

$$\boldsymbol{x}(t + \Delta t) = \boldsymbol{x}(t) + \Delta t \cdot \dot{\boldsymbol{x}}(t). \quad (10)$$

The Gaussian white noise model effectively models the real-world noise impact on the control system of the manipulator. It incorporates a moving average filter to mitigate noise effects, enhancing control accuracy and reliability in noisy environments.

## V. SIMULATION EXPERIMENT

In this section, the simulation is implemented based on physical structure parameters of the 7-DOF manipulators KUKA LBR iiwa 7 R800, thus verifying the accuracy and feasibility of the proposed double-indicator scheme based on the DNN solver (6). In addition, comparing it with the conventional single-objective scheme, the validity of the simple anti-noise mechanism is verified in the simulation(9).

### A. Parameter Setting

In the simulation, the MKE scheme weight is set as $\xi = 0.3$, CMG scheme modulation factor is set as $m = 10.2$, position error amplification factor is set as $\delta = 10.2$, convergence coefficient is set as $\rho = 9 \times 10^{-5}$, and $G$ is set as $I$. In addition, for the convenience of the simulation, the initial joint angles are set as $\boldsymbol{\theta}(0) = (-2.357, -0.547, 0.023, 1.721, 0.005, -0.681, 0)^{\text{T}}$ rad, the simulation time $t = 10$ s, the trajectory planning interval is $0.0001$ s, the sliding average window size

$N = 10$, the white noise amplitude is 0.01, and the pre-defined heart-shaped trajectory is tracked based on the simulation of the KUKA LBR iiwa 7 R800 manipulator. The generation formula of the heart-shaped trajectory is set as

$$\begin{cases} x(t) & = x(0) + a \cdot 16 \cdot \sin^3(\omega t) \\ y(t) & = y(0) + a \cdot (13\cos(\omega t) - 5\cos(2\omega t) - 2\cos(3\omega t) \\ & \quad - \cos(4\omega t)) - 5a \\ z(t) & = z(0). \end{cases}$$

### B. Comparison And Result Analysis

Based on the tracking simulation of the intended trajectories, the above scheme combining the DNN solver (6) and the simple anti-noise mechanism (9) shows its feasibility in controlling the redundant manipulators. In Fig. 1(d), the end-effector position deviations stay below $1 \times 10^{-4}$ m throughout the simulation, indicating a high tracking accuracy. In Fig. 1(b), the joint angles of the manipulators are kept within their physical limits, i.e., $\underline{\theta} = -2.5$rad, $\overline{\theta} = 2.5$rad, indicating that the physical constraints are taken into account, thus confirming the practical feasibility of the control scheme. The curves in Fig. 1(c) depicts the variation of angular velocity under the proposed control scheme, showing that the angular velocity variations of all joints are within the specified range and the transition is smooth, reflecting the high stability of the proposed control scheme.

The research results indicate that the innovative control scheme based on DNN exhibits excellent performance in trajectory tracking tasks of the manipulator. This approach achieves high precision and stability and significantly reduces the energy consumption, demonstrating the potential as a comprehensive control strategy.

The core of this control scheme lies in the double-indicator optimization framework, which is implemented via a DNN solver. A vital feature of this framework is the flexibility, allowing researchers to balance the importance of different performance metrics by adjusting the objective weights $\xi$. Simulations conducted in a noiseless simulation environment demonstrate the adaptability of this method in Table I.

Overall, this research not only advances the technological development in the field of the manipulator control but also offers valuable insights into achieving multi-objective optimization of complex systems.

### C. Comparisons

The proposed double-indicator scheme for DNN-based solutions (6) focuses on the objectives by adjusting the objective weight values $\xi$, and Table I demonstrates the multifaceted performance of the mentioned scheme for a series of different values of $\xi$ without the inclusion of simulated noise. The comparison results indicate that the highest accuracy in the simulations is achieved when $\xi = 0.9$, suggesting that when the weight of the MKE scheme is higher, the value of MAX $E$ decreases, leading to improved stability and accuracy in the control process. When the weight of the CMG scheme is relatively more significant, i.e., when $\xi$ is smaller, the value of MAX $|\theta(T) - \theta(0)|$ decreases accordingly, resulting in a

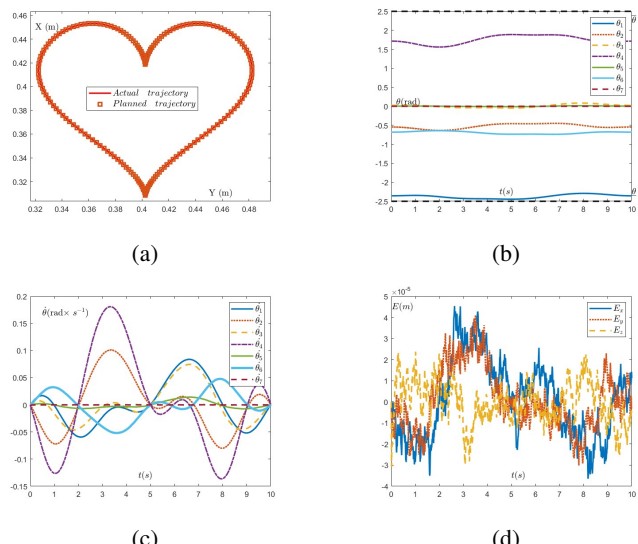

Fig. 1: Simulation results of KUKA manipulators with heart-shaped tracking trajectory analog noise added executed by the double-indicator scheme based on DNN solver (6). (a) Target versus actual trajectories. (b) Time histories of joint angles. (c) Time histories of joint velocities. (d) Time histories of end-effector spatial position error.

minor variation in the joint angle from the start to the end of the trajectory. Moreover, considering objectives and accuracy, $\xi = 0.2$ represents an optimal balance between the schemes.

For the effect of the noise incorporation and the proof of the effectiveness of the anti-noise mechanism, Fig. 2 shows the comparison of the error curves for the three cases of no noise added, noise added, and noise added with an anti-noise mechanism for the $\xi = 0.2$ scenario.

Table II compares the errors of three models. Due to the nature of the trajectories, it is easy to observe that the errors are significant at the beginning, and the maximum error (MAX $E(10^{-5})$) is mainly determined by this. It is less affected by the slight amplitude noise. In addition, the complete picture of the overall error may not be given; thus, the mean squared error (MSE) is also compared.

Fig. 2(a) shows the scenario without the added noise. The error curves ($E_x$, $E_y$, $E_z$) are relatively smooth and have small amplitudes, which indicates that the system can maintain high accuracy without the noise interference. Fig. 2(b) illustrates the situation after the addition of the voice noise. As the amplitude increases significantly, the error curve becomes more unstable, which indicates that the noise causes a large disturbance to the system, resulting in a significant increase in the error. Fig. 2(d) presents the case after adding noise and applying an anti-noise mechanism. Compared to Fig. 2(b), the fluctuation and amplitude of the error curve are reduced, which indicates that the anti-noise mechanism reduces the interference of noise on the system to some extent and improves the system's noise immunity. Comparing Fig. 2(a) and (c), the impact of incorporating the noise reduction mechanism on the error is

TABLE I: Performance Metrics for Different $\xi$ Values

| $\xi$ | 0.1 | 0.2 | 0.4 | 0.5 | 0.6 | 0.8 | 0.9 |
|---|---|---|---|---|---|---|---|
| MAX $E(10^{-5})$ | 7.25 | 5.59 | 6.50 | 4.06 | 7.88 | 4.24 | 1.75 |
| MAX $|\theta(\mathrm{T}) - \theta(0)|\,(10^{-2})$ | 1.18 | 1.74 | 3.88 | 5.88 | 8.71 | 27.0 | 219 |
| MAX $\dot{\theta}^{\mathrm{T}} G \dot{\theta}(10^{-6})$ | 4.54 | 4.54 | 4.54 | 4.54 | 4.54 | 4.54 | 4.54 |

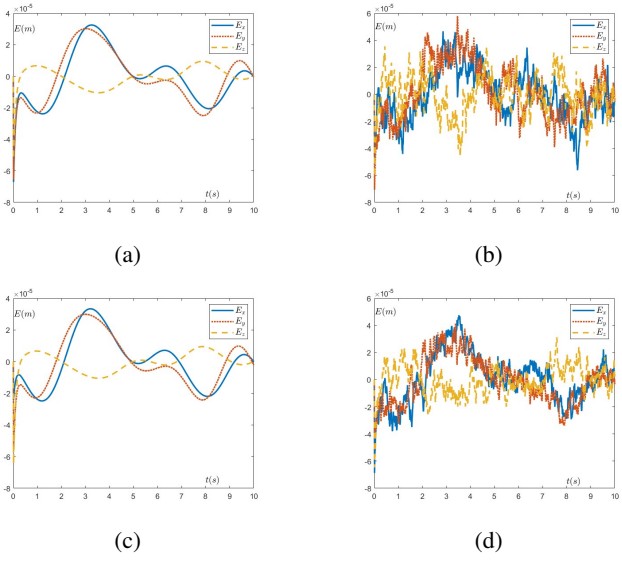

(a)  (b)

(c)  (d)

Fig. 2: Simulation results of the double-indicator scheme based on the DNN solver (6) based on the simulation results of the KUKA manipulators with a heart-shaped tracking trajectory($\xi = 0.2$). (a) Time course of end-effector position error without adding noise. (b) Time course of end-effector position error after adding noise without adding an anti-noise mechanism. (c) Time course of end-effector position error after adding anti-noise mechanism but without noise. (d) Time course of end-effector position error after adding noise and an anti-noise mechanism.

TABLE II: Comparison of Errors in Different Scenarios

| Scenario | MAX $E$ | MSE |
|---|---|---|
| **Without Noise** | | |
| No Filter | $6.70 \times 10^{-5}$ | $2.00 \times 10^{-10}$ |
| Filtered | $6.47 \times 10^{-5}$ | $1.97 \times 10^{-10}$ |
| **With Noise** | | |
| No Filter | $7.07 \times 10^{-5}$ | $3.28 \times 10^{-10}$ |
| Filtered | $6.87 \times 10^{-5}$ | $2.49 \times 10^{-10}$ |

not apparent when no noise is added. However, the noise reduction mechanism still significantly reduces the system error at this time, which can be seen in Table II.

The simulation outcomes demonstrate that, in the absence of noise, the system exhibits a maximum error (MAX $E$) of $5.63 \times 10^{-5}$ meters and a mean square error (MSE) of $1.94 \times 10^{-10}$ meters. After introducing unfiltered noise, the maximum error escalates to $7.75 \times 10^{-5}$ meters, while the mean square error rises to $3.42 \times 10^{-10}$ meters. These findings underscore the detrimental effect of noise on the system's accuracy.

Upon implementation of the filtering mechanism, a significant reduction in the maximum error to $6.71 \times 10^{-5}$ meters and a decrease in the mean square error to $2.67 \times 10^{-10}$ meters are observed. Even without noise, the filtering mechanism reduces the system error. These results substantiate the efficacy of the anti-noise mechanism in mitigating error propagation. While the error levels with the filter in place remain elevated compared to the noise-free scenario, the marked improvement over the unfiltered noise condition unequivocally demonstrates the effectiveness and necessity of the noise mitigation strategy.

### D. Physical Simulation

To demonstrate the operability and realism of the proposed scheme, physical simulations based on the KUKA LBR iiwa 7 R800 manipulators using CoppeliaSim analog simulation software are carried out, where the trajectories of the end-effector were recorded, and some central time nodes are extracted and displayed in Fig. 3.

Fig. 3 clearly shows the excellent tracking performance of the manipulator along the cardioid trajectory. Specifically, the manipulator accurately follows the predefined cardioid trajectory with the tracking error kept to a minimum value. It highlights the high accuracy and stability of the proposed control scheme. In addition, it validates the effectiveness and practicality of the redundant manipulator control scheme in complex trajectory-tracking tasks.

## VI. CONCLUSION

This paper has proposed a new double-indicator scheme for trajectory tracking of redundant manipulators. The scheme considers the position of the end effector and kinetic energy consumption while introducing noise-resistant mechanisms to enhance accuracy and stability further. Additionally, a DNN solver, validated to exhibit global exponential convergence when handling optimization problems derived from this control scheme, has been employed to achieve fast and efficient control of the manipulators. Finally, this scheme's effectiveness, high reliability, and specificity in the control of redundant manipulators have been demonstrated through comparative simulations.

## ACKNOWLEDGMENT

This paper is supported by the Research Capability Improvement "Project of Central Universities Basic Scientific Research Business Expenses Special Fund" of Lanzhou University in 2024 under the title of "Multi-Robot Competitive Coordination Planning Based on the Winner-Take-All Strategy" (Project No. lzujbky-2024-ou02).

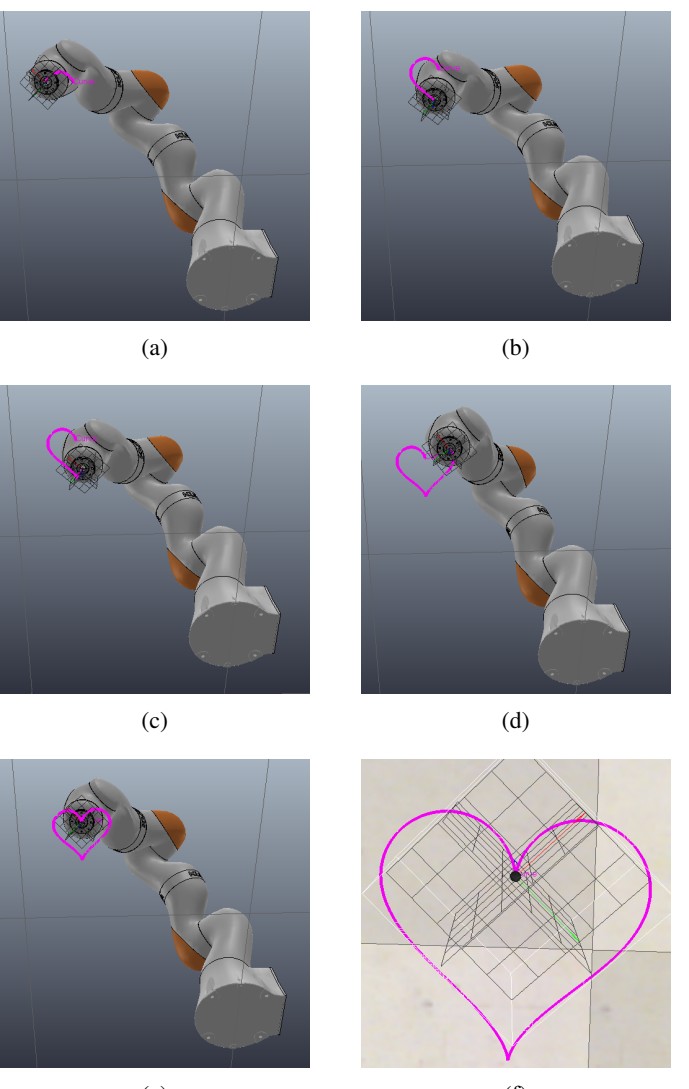

(a)            (b)

(c)            (d)

(e)            (f)

Fig. 3: CoppeliaSim-based simulation results of DNN solver-based double-indicator scheme controlling KUKA 7-degree-of-freedom manipulators at different time nodes, tracking predefined heart-shaped trajectories. (a) $t = 2$s. (b) $t = 4$s. (c) $t = 6$s. (d) $t = 8$s. (e) $t = 10$s. (f) Heart-shaped trajectory tracking results.

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
