# OpenReview forum: "Enhanced Orientation Tracking for Redundant Manipulators via DNN-Based Double Control"
_IEEE.org/ICIST/2024/Conference — IEEE ICIST 2024 Conference Submission_

### Official Review · Reviewer_ugCW · 2024-08-27
**Enhanced Orientation Tracking for Redundant Manipulators via DNN-Based Double Control**

**Rating:** 7
**Confidence:** 4

**Review:**

In this paper,  a new double-indicator control scheme of redundant manipulators is proposed. This work is well organized and novel. Below are some comments. (1) The contributions should be illustrated in a clearer manner. For example, what is the main improvement of the paper compared to the existing results. The authors should explain the unique contributions of this paper. (2) The simulation results should be explained more carefully. (3) The paper is well presented, spelled correctly. I recommend authors to carefully read the entire paper to find possible misspellings.

---

### Official Review · Reviewer_szNk · 2024-08-27
**Acceptable**

**Rating:** 7
**Confidence:** 3

**Review:**

The paper presents a novel control scheme for redundant manipulators using a dynamic neural network (DNN)-based double-indicator control approach. This topic is interesting, the following comments need to further consider: 1. Although the paper briefly compares the proposed method with traditional single-objective schemes, a more detailed comparison with other state-of-the-art methods is necessary. This would help to better position the proposed approach within the existing body of knowledge and highlight its unique contributions. 2. Providing more detailed implementation guidelines or a case study on how to apply the proposed method to different types of manipulators could enhance the paper's usefulness for practitioners.3.The format of the references should be unified.

---

### Official Review · Reviewer_Rota · 2024-08-27
**This article is very interesting and a good one.**

**Rating:** 7
**Confidence:** 5

**Review:**

In this paper,  a new double-indicator scheme for trajectory tracking of redundant manipulators has been proposed. The obtained result is valuable and can be accepted if the following problems can be clarified.
1. The main contributions and significance should be rewritten to illustrate the main ideas and main work of this paper.
2. This paper uses an untrained dynamic neural network solver. What are the advantages of this solver?
3. The format of references needs to be uniform.
4. The quality of language needs significant improvement, and professional editing may be necessary.

---

### Decision · Program_Chairs · 2024-09-08

Accept (Oral)